

# Exposure to *Batrachochytrium dendrobatidis* metabolites altered ghost shrimp behavior and reduced mortality

Ellisa Carla Parker-Athill, Liam C. Muldro, Aiza J. Malinias and
Taegan A. McMahon

Department of Biology, Connecticut College, New London, Connecticut, United States

## ABSTRACT

*Batrachochytrium dendrobatidis*, or *Bd*, is a waterborne, pathogenic chytrid fungus implicated in the decline and extinction of hundreds of amphibian species worldwide. *Bd* can infect multiple taxa, causing disease in susceptible species associated with epidermal degradation, lethargy, weight loss, gill damage, and disruption of osmoregulation and cardiac dysfunction, ultimately leading to mortality in many instances. *Bd* produces water soluble chemicals (*Bd* metabolites) that, while implicated in infection and *Bd*-related pathology, have also been shown to have prophylactic effects for amphibians. This study examined the impact of *Bd* metabolite exposure on critical non-target freshwater invertebrates to better understand how *Bd* metabolites might impact non-target species if used as a prophylaxis in the field. We examined the effects of *Bd* metabolites on the freshwater species, *Palaemonetes paludosus*, or ghost shrimp, focusing specifically on the effects on behavior, cardiac function, and mortality. Shrimp were randomly split into two treatment groups and on day 4, they were dosed with 1 ml of either an artificial spring water (ASW) control or *Bd* metabolite treatment. We found that ghost shrimp exposed to *Bd* metabolites experienced decreased heart rate, reduced threat response behaviors, and reduced mortality, when compared to those exposed to an ASW control. Our findings suggest that exposure to the *Bd* metabolites may actually provide some benefits to ghost shrimp as it decreased mortality compared to controls. While more research is needed to understand if there are indirect impacts of the metabolites associated with the behavioral and cardiac changes, this research highlights that exposure to *Bd* metabolites does not appear to be detrimental to these critical, keystone freshwater invertebrates. This is particularly important given that *Bd* metabolites may be used prophylactically in the field, which may lead to non-target organism exposure.

Corresponding author
Ellisa Carla Parker-Athill,
eparkerat@conncoll.edu

## INTRODUCTION

*Batrachochytrium dendrobatidis*, also referred to as *Bd*, is a pathogenic chytrid fungus implicated in the decline and extinction of hundreds of amphibian species worldwide (*Scheele et al., 2019*). This water borne fungus reproduces *via* sporangia, and is spread by

the dispersion of infectious, flagellated zoospores (*Longcore, Pessier & Nichols, 1999*). Infections may be transmitted *via* direct contact with an infected host or with the free-swimming mobile zoospores. Indirect exposure is thought to occur through contact with contaminated environmental reservoirs (*e.g.*, sediment (*Kirshtein et al., 2007*), boulders (*Longcore, Pessier & Nichols, 1999*; *Wixson & Rogers, 2009*), bromeliads (*Cossel & Lindquist, 2009*), aquatic birds (*Garmyn et al., 2012*; *Burrowes & De la Riva, 2017*) or snakes (*Kilburn, Ibáñez & Green, 2011*)), although transmission through this route has not been confirmed.

In amphibians, *Bd* infection can impact the keratinized layer of the epidermis and can cause pathological features such as skin degradation, sloughing, hyperkeratosis, hyperplasia, ulceration, erosions, and necrosis (*Berger et al., 1998*; *Grogan et al., 2018*). Infected amphibians may also show evidence of lethargy, loss of appetite, behavioral changes and in many instances, mortality (*Gabor, Fisher & Bosch, 2015*). The epidermis is critically important to physiological processes such as osmoregulation, thermoregulation, gas exchange and indirectly cardiac function. *Bd* infections have been directly implicated in the disruption of osmoregulation, including imbalances in sodium and potassium ion concentrations (*Peterson et al., 2013*; *Voyles et al., 2009*) and cardiac dysfunction, including decreased stroke volume, delayed cardiac cycles (*Salla et al., 2018*) and asystolic cardiac arrest (*Voyles et al., 2009*), any of which can lead to mortality.

*Bd* metabolites, water soluble chemicals produced by the chytrid fungus, have also been shown to cause pathological outcomes in amphibians. While the exact chemical composition of *Bd* metabolites is still unknown, we know that *Bd* produces proteolytic enzymes which have been implicated in skin disorders. Other metabolites have been shown to have immunomodulatory and inhibitory potential (*e.g.*, methylthioadenosine, tryptophan, and polyamine spermidine; *Rollins-Smith, Reinert & Burrowes, 2015*; *Rollins-Smith et al., 2019*). Interestingly, *Bd* metabolites have also been shown to produce prophylactic effects, although the exact metabolites responsible have not been identified. This may be because the composition of metabolites *Bd* produces changes depending on the local environment (*Rollins-Smith, Reinert & Burrowes, 2015*). Specific *Bd* metabolites may have different impacts, some detrimental and others potentially prophylactic. Differences in the concentration of *Bd* metabolites an organism is exposed to may also account for the range of impacts, from toxic to pharmaceutically beneficial. While exposure to high doses of *Bd* metabolites may be detrimental to some freshwater species (*McMahon et al., 2013*; *Nordheim, Grim & McMahon, 2021*), studies have shown that exposure to low doses of *Bd* metabolites may actually produce beneficial effects. Indeed, amphibians exposed to *Bd* metabolites responded to this prophylactic treatment exposure, and had reduced subsequent *Bd* infection loads, and in some cases, there was even lower infection prevalence (*McMahon et al., 2014*, *2023*, *2024*; *Barnett et al., 2021*, *2023*; *Nordheim et al., 2022*). Additionally, tadpoles exposed to low concentrations of *Bd* metabolites experienced higher developmental speeds, while maintaining equal growth and survival compared to control tadpoles (*McMahon, Laggan & Hill, 2019*). These findings suggest that while high concentrations of *Bd* metabolites may impact important physiological processes such as gas exchange and osmoregulation, lower concentrations

may be neutral or may even present a prophylactic potential against *Bd* infection, including reduced pathogen load, increased tolerance or resistance and potentially reducing adverse outcomes such as disease severity and mortality.

The concept that *Bd* metabolites may be beneficial is not entirely novel. Other fungal species have been shown to produce metabolites with pharmaceutically beneficial effects for freshwater organisms. The fungal species *Aspergillus niger*, for example, has been shown to increase survival in white shrimp (*Zhang et al., 2023*) in aquaculture. In fact, *A. niger* has been shown to enhance growth, immunity and disease resistance (*Cheng et al., 2023*). Additionally, fungal polysaccharides have been found to yield immunostimulation, increased pathogen resistance and even improve growth in crustaceans (*Mohan et al., 2019*). Indeed, many fungal species have been shown to produce bioactive metabolites with antimicrobial, including antifungal properties, that have proved critical to the aquaculture industry (*Xu et al., 2021*; *Du et al., 2022*).

In this study, we examined the effects of *Bd* metabolites on the freshwater species, *Palaemonetes paludosus*, more commonly known as ghost shrimp. It is important to note that despite *Bd*'s prevalence within amphibian populations, it is not exclusive to amphibians as it has been identified on non-amphibian organisms, such as wading birds (*Garmyn et al., 2012*) and snakes (*Kilburn, Ibáñez & Green, 2011*), and confirmed on museum specimens of aquatic birds originating from the Bolivian Andes (*Burrowes & De la Riva, 2017*). In its non-amphibian hosts, specifically crayfish, *Bd* has been identified growing in the gastrointestinal tract and on the exoskeleton (*McMahon et al., 2013*; *Brannelly et al., 2015*; *Román, O'Neil & James, 2016*; *Oficialdegui et al., 2019*). *Bd* related mortality in these freshwater invertebrates is likely due to gill damage and the resulting impairment in gas exchange (*McMahon et al., 2021*; *Nordheim, Grim & McMahon, 2021*), and potentially osmoregulation, given the role the gills play in these processes. Similarly, *Bd* metabolites have been shown to cause pathological outcomes in non-amphibian organisms. Research has demonstrated that gill damage and mortality can occur in crayfish exposed to high concentrations of *Bd* metabolites, and that these detrimental impacts occur even in the absence of *Bd* infection (*McMahon et al., 2013*; *Nordheim, Grim & McMahon, 2021*). It should be noted that these detrimental effects were not seen when crayfish were exposed to low concentrations of *Bd* metabolites, suggesting that not only might specific metabolites confer prophylactic benefits, but that a concentration dependent relationship between the prophylactic and pathological effects of *Bd* metabolites may exist (*McMahon et al., 2013*; *Nordheim, Grim & McMahon, 2021*).

Ghost shrimp are native to the southeastern coastal plains of North America and reside in shallow regions of freshwater ponds, lakes, and streams. These small invertebrates are scavengers and bioturbators, and they co-occur with *Bd* and amphibians. Through borrowing activities, ghost shrimp can alter carbon and nutrient cycling, and influence microbiota interactions and dispersal. They are a critical freshwater species, making them an ideal model to understand the impact of *Bd* metabolites and indirect exposure to *Bd* on freshwater organisms (*Wada, Urakawa & Tamaki, 2016*; *Adhikary, Klerks & Chistoserdov, 2024*). Additionally, ghost shrimp are ideal for studying bioactive compounds because their characteristic behavioral responses to threats, *e.g.*, hiding and escape responses, offer

strong opportunities to understand the effects of *Bd* metabolites on well-studied critical behaviors. The biochemical pathways, including those that regulate these behaviors, have also been characterized extensively in crayfish, which share neurochemical homology to ghost shrimp (*Sosa et al., 2004*; *Wongprasert et al., 2006*; *Fossat et al., 2014*). Finally, ghost shrimp's characteristic translucent carapace and bodies allow for unpresented physical assessment, including monitoring of heart rate and changes to the physical appearance of the carapace.

To date, we know little about the direct and indirect effects of *Bd* on non-amphibian organisms. Indeed, most research on *Bd* has focused on amphibian conservation, which is not surprising given the severe decline in this taxon. The goal of this study was to better understand the effects of *Bd* metabolites on non-amphibian, freshwater organisms, which is particularly critical if *Bd* metabolites are to be used as a prophylactic at a community level. Additionally, understanding the complex dynamics of indirect exposure to *Bd* on co-occurring freshwater invertebrates may provide important insight into the mechanisms through which *Bd* is affecting population dynamics, and will provide a more holistic view of how *Bd* impacts the freshwater ecosystems.

## MATERIALS AND METHODS

### Animal husbandry

Ghost shrimp (*Palaemonetes paludosus*) obtained from Northeast Brine Shrimp, LLC (Marietta, GA, USA) were individually housed in plastic containers containing 800 ml fresh artificial spring water (ASW; *Cohen, Neimark & Eveland, 1980*), with a single aeration stone and gentle aeration. Half of the tank was covered on all sides including the lid, to create a dark and light environment within the same container. Shrimp were maintained on a 12:12 h light/dark cycle at ~21 °C and they were fed every other day with high quality shrimp food (Hikari Sales USA Inc., Hayward, CA, USA). The uneaten food was removed daily to avoid water fowling and cross contamination was avoided. Shrimp were checked daily for mortality, molting, and overall health status, and visibly distressed (laying on back, pale white color) or dead animals were promptly removed and processed for future analysis.

### Treatment and experimental procedures

To verify that the shrimp were *Bd* negative, we swabbed the carapace of each shrimp 10 times and made sure to swab the entire surface of the carapace. These swabs were frozen in a lab grade −20 °C freezer and were processed for presence of *Bd* with quantitative polymerase chain reaction (qPCR; see below for qPCR methods). All shrimp were *Bd* negative prior to the start of this experiment. Baseline measurements of heart rate (see below for heart rate analysis methods), and total extended length of shrimp (mm) were recorded prior to the start of the experiment.

Prior to dosing, referred to hereafter as pre-exposure, shrimp were allowed to acclimate in their container for 3 days and shrimp were not removed from this container during the experimental trials. As a baseline for our behavioral monitoring, we conducted daily pre-exposure behavioral assays noting shrimp: activity, position in the container (light *vs*

dark side of container), and height of shrimp in the water column for 3 days prior to dosing (see *Behavioral monitoring* for methods). Shrimp were randomly split into two treatment groups (*n* = 20 shrimp/treatment). On day 4, shrimp were dosed with 1 ml of either an ASW control or *Bd* metabolite treatment (see below for *Bd culture* and *Bd metabolite prophylaxis preparation* methods). Post exposure behavioral testing was completed on days 4–14, and all remaining shrimp were euthanized by flash freezing and processed on post exposure day 14.

## Heart rate analysis

A three-second video was taken of the heart of all living shrimp at the beginning and end of the experiment. Ghost shrimp are mostly translucent and their heart is clearly visible through their carapace. The pre-exposure heart rate was determined for all shrimp and the post-exposure heart rate was determined for all shrimp that survived to the end of the experiment.

## *Bd* culture and *Bd* metabolite prophylaxis preparation

*Bd* (strain JEL 419; strain isolated in Panamá during an amphibian decline event) was cultured on 1% tryptone agar plates for 14 days at 17 °C. These cultures were flooded with ASW for 10 min and then the ASW and *Bd* from all plates were homogenized to create the *Bd* positive stock ($1 \times 10^5$ zoospores/ml). We used the same method to create the *Bd* metabolite prophylaxis here as our previous amphibian exposure work (please see *Nordheim et al., 2022* for detailed methods). In brief, to create the *Bd* metabolite prophylaxis for this study, the *Bd* positive stock was frozen, thawed and then passed through a 1.2 μm filter (GE Whatman Laboratory Products, Kent, UK) to remove all the zoospores and zoosporangia. This remaining homogenate contained just the water-soluble metabolites that the *Bd* produced while suspended in the stock (*Bd* metabolite prophylaxis: $1 \times 10^5$ zoospores removed/ml). We visually verified there was no viable *Bd* left in the *Bd* metabolite prophylaxis with a compound microscope. Additionally, we plated the *Bd* metabolite prophylaxis on 1% tryptone agar plates (*n* = 3 plates) and monitored for growth for 14 days. There was no *Bd* growth. The *Bd* metabolite concentration was chosen because it was found to negatively impact crayfish (*McMahon et al., 2013*; *Nordheim, Grim & McMahon, 2021*) and therefore might be in the range that could impact ghost shrimp as well. Additionally, it was higher than the effective prophylaxis concentration (*McMahon et al., 2024*) and so this would give us a strong chance of seeing deleterious impacts if there are some.

## Behavioral monitoring

Behavioral screening was performed daily pre-exposure (during the 3-day acclimation period to establish a baseline and to account for individual variation), and post-exposure (to assess the effects of *Bd* metabolites on ghost shrimp behavior and activity level). All behavioral screening was performed in the morning during the light cycle, in the shrimps' home tank and it occurred without disturbing the shrimp.

*Activity monitoring*: Ghost shrimp activity level was scored on a three-point scale, with 1 = no activity, 2 = mild activity, 3 = swimming actively. Ghost shrimp height was recorded as distance (cm) from the bottom of the container.

*Modified light dark box test*: Briefly, to assess whether ghost shrimp were found on the light or dark side of the tank, a modified light dark box was created. The shrimp tank, or chamber, was made of a clear plastic, which allowed light to pass through. One half of the chamber remained uncovered, and the other half was fully covered with dark construction paper on the sides, bottom and top. This set up created a shrimp tank that allowed light on one side and was dark on the other. We identified whether the ghost shrimp were on the dark or light side of the chamber daily during the pre- and post-exposure sections of the experiment.

*Escape response test*: Four days post-exposure, each shrimp's escape response was measured by physically touching the telson of the ghost shrimp with a sterile plastic inoculation loop to determine whether each shrimp exhibited an evasive movement in response to a stimulus.

## Gill damage

Gill damage (gill tissue recession) was measured following the protocol established for crayfish (see *McMahon et al., 2013* and *Nordheim, Grim & McMahon, 2021*). In brief, ghost shrimp were frozen and then thawed prior to dissection and gill extraction. A large section of gill was removed from each ghost shrimp to ensure there was ample undamaged gill tissue to photograph, and photographs of each gill were taken with a compound light microscope at 10× magnification. ImageJ was used to collect measurements from each gill and all measurements were collected double-blind. We measured the distance between the tip of the external gill cuticle to the hemolymph and gill tissue ($n = 3$ measurements/gill).

## qPCR methods

The qPCR protocol proposed by *Boyle et al. (2004)* was followed to verify the ghost shrimp were *Bd* negative prior to the start of the experiment. Briefly, DNA was extracted from the swabs using PrepMan Ultra (Applied Biosystems, Foster City, CA, USA) and TaqMan Exogenous Internal Positive Control Reagents (Applied Biosystems, Foster City, CA, USA) were used to verify there was no inhibition; there was no evidence of inhibition. All samples were run on one 96 well plate and the standard curve ranged from 0.84–840 genome equivalents (GE; standards were *Bd/Bsal* plasmid standards purchased from Pisces Molecular). All ghost shrimp were *Bd* negative at the beginning of the experiment (we have never received *Bd* infected ghost shrimp from this company).

## Statistical analysis

All analyses were conducted in R using R studio (version 4.2.2; *R Core Team, 2022*). A general linear model (package: stats, function: glm, family: gaussian; *R Core Team, 2022*) was used to verify that there was no difference in initial shrimp length between the treatment groups prior to the start of the experiment. A generalized linear model (package: glmmTMB, function: glmmTMB, family: gaussian; *Brooks et al., 2017*) was used to

determine if there was an effect of treatment on whether ghost shrimp spent more time on the dark side of the enclosure post treatment exposure. Time spent on the dark side pre-exposure, number of observations post-exposure (number of times behavioral observations were made for each individual after exposure; this number varied due to mortality), and mortality status were accounted for as covariates in the model.

A general linear model (package: stats, function: glm, family: gaussian) was used to determine whether there was an effect of treatment or time after exposure on gill recession, whether there was a correlation between gill recession and activity level, whether there was an effect of treatment or number of post exposure observations on post exposure activity level, whether there was an effect of treatment or number of post exposure observations on height post exposure (taking into consideration average height pre-exposure), and whether there was an effect of treatment or number of post exposure observations on total number of molts post exposure. A general linear model (package: stats, function: glm, family: gaussian) was also used to determine if there was a difference in initial heart rate between the treatment groups prior to exposure to the treatments and to determine whether there was an effect of treatment on final heart rate (taking into consideration the initial heart rate of each shrimp). We used a general linear model (package: stats, function: glm, family: gaussian) to determine if treatment had an effect on whether shrimp exhibited an evasive movement in response to a stimulus. A survival analysis was performed and the data was fit to a cox proportional hazards regression (package: survival, function: coxph; *Therneau, 2024*) to determine if there was an effect of treatment on mortality.

Figures 1–3 were created in R (Fig. 1: package: survminer, function: ggsurvplot (*Kassambara, Kosinski & Biecek, 2024*)); Figs. 2 and 3: package: ggplot2, function: ggplot; *Wickham, 2016*, and Fig. 4 was created in Excel (Version 16.78.3).

## RESULTS

### *Bd* metabolites reduced mortality in ghost shrimp

There was reduced mortality in ghost shrimp exposed to the *Bd* metabolite treatment compared to the ASW control ($z = -2.23$, $p = 0.026$; Fig. 1). Ghost shrimp exposed to the *Bd* metabolite treatment had 48.4% lower mortality than those exposed to the ASW control (95% Confidence Interval (CI) for mortality proportion for ASW control: 0.454 and 0.883, and *Bd* metabolites: 0.082 and 0.503, lower and upper limit respectively; calculated using a method described by *Wilson, 1927*).

### *Bd* metabolites reduced heartrate but did not induce gill recession in ghost shrimp

Ghost shrimp heart rate was measured over a 3 s period pre and post exposure to *Bd* metabolites. Ghost shrimp exposed to the *Bd* metabolites had reduced heart rates compared to those exposed to the ASW controls 8 days after exposure ($t = -2.56$, $p = 0.017$; Fig. 2). Ghost shrimp exposed to the *Bd* metabolite treatment had 19.57% lower heart rates than those exposed to the ASW control. There was no difference in initial heart rate

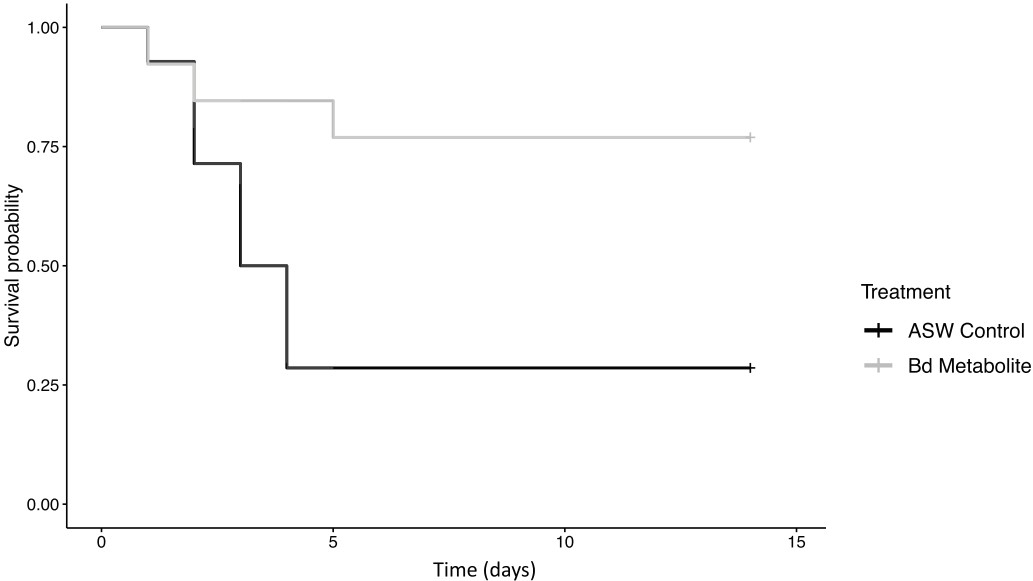

**Figure 1 *Bd* metabolites reduced mortality in ghost shrimp.** Ghost shrimp (*Palaemonetes paludosus*) exposed to the *Batrachochytrium dendrobatidis* (*Bd*) metabolite treatment had increased survival compared to those exposed to an ASW control (*n* = 20 shrimp/treatment). The Kaplan-Meier curve was generated in RStudio.

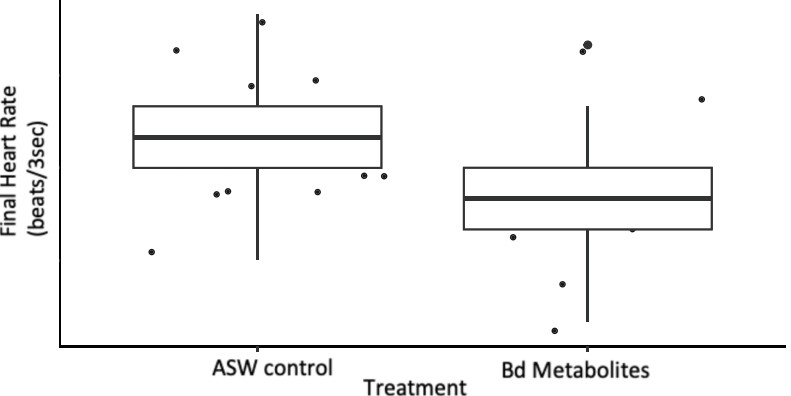

**Figure 2 *Bd* metabolites reduced heartrate in ghost shrimp.** Ghost shrimp (*Palaemonetes paludosus*) exposed to the *Batrachochytrium dendrobatidis* (*Bd*) metabolite treatment had a reduced final heart rate compared to those exposed to an ASW control (*n* = 20 shrimp/treatment). The top and bottom of the box represents the 75th and 25th percentiles, respectively, the line = median, the x = mean, and the whiskers = the minimum and maximum values.

between the treatment groups (t = −1.39, *p* = 0.176), however there was a positive correlation between initial heart rate and final heart rate (t = 2.87, *p* = 0.008).

There was no effect of treatment on gill recession (t = 1.27, *p* = 0.217) and this did not change with time after exposure (t = 0.343, *p* = 0.734). There was also no correlation between gill recession and activity level (t = −0.703, *p* = 0.489).

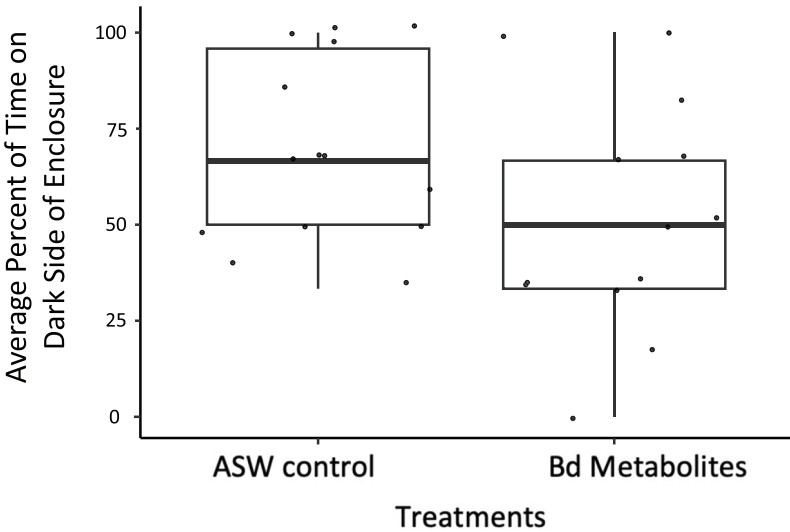

**Figure 3 Ghost shrimp exposed to the *Bd* metabolites spent less time on the dark side of the enclosure post exposure.** Ghost shrimp (*Palaemonetes paludosus*) exposed to the *Batrachochytrium dendrobatidis* (*Bd*) metabolite treatment spent less time on the dark side of the enclosure post exposure compared to those exposed to the ASW control (*n* = 20 shrimp/treatment). The top and bottom of the box represents the 75th and 25th percentiles, respectively, the line = median, the x = mean, and the whiskers = the minimum and maximum values.

## *Bd* metabolites modulate ghost shrimp threat based behavioral responses

*Bd* metabolites did have an effect on ghost shrimp behaviors. Ghost shrimp exposed to the *Bd* metabolites spent 17.77% less time on the dark side of the enclosure post exposure compared to those exposed to the ASW control ($z = -1.965$, $p = 0.0494$; Fig. 3). The amount of time spent on the dark side pre-exposure, the number of observations made post-exposure, and mortality status did not have an effect on time spent on dark side of the container post-exposure ($z = 3.95$, $p = 0.693$, $z = -1.172$, $p = 0.241$; $z = -1.442$, $p = 0.149$, respectively).

Ghost shrimp exposed to the *Bd* metabolites were 70% less likely to exhibit evasive movements 8 days after exposure when stimulated compared to ghost shrimp exposed to the ASW control ($t = -2.82$, $p = 0.015$; Fig. 4). There was no effect of treatment or number of post exposure observations on post exposure activity level ($t = -0.394$, $p = 0.697$, $t = 1.68$, $p = 0.107$, respectively).

There was no effect of treatment, average pre-exposure height or number of post exposure observations on post exposure height ($t = -0.321$, $p = 0.751$, $t = 0.371$, $p = 0.715$, $t = 1.35$, $p = 0.191$, respectively). There was no effect of treatment on the total number of molts post-exposure ($t = -1.62$, $p = 0.12$), but there was an impact of the number of post exposure observations on the number of molts ($t = 3.48$, $p = 0.002$). There was no difference in ghost shrimp length between the two treatment groups at the beginning of the experiment ($t = 1.516$, $p = 0.138$).

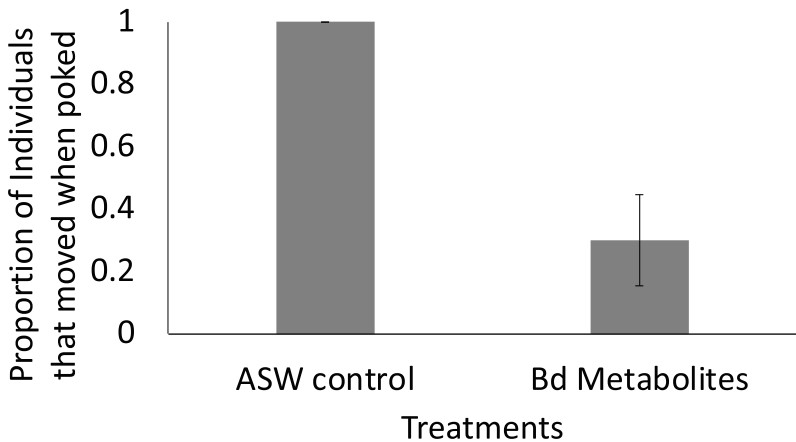

**Figure 4 Fewer ghost shrimp exposed to the *Bd* metabolites exhibited evasive movements compared to those exposed to the ASW control.** Fewer ghost shrimp (*Palaemonetes paludosus*) exposed to the *Batrachochytrium dendrobatidis* (*Bd*) metabolite treatment exhibited evasive movements when stimulated on the telson 8 days after exposure compared to ghost shrimp exposed to the ASW control ($n = 20$ shrimp/treatment). The proportion of the overall population of ghost shrimp that exhibited an escape response ± SE is shown.

## DISCUSSION

This study found that exposure to *Bd* metabolites altered ghost shrimp behavior and mortality, but not necessarily in an expected manner. We found that ghost shrimp exposed to *Bd* metabolites experienced decreased mortality when compared to those exposed to ASW, suggesting that *Bd* metabolites may actually improve survival. The mechanism for increased survival in ghost shrimp was not studied here, but previous research found that *Bd* metabolites altered the host microbiome, shifting the microbiome towards more protective biota (*Siomko et al., 2023*). The microbiome, particularly the gut microbiome, has been shown to be intrinsically related to immune function (*Holt et al., 2021*). It is certainly possible that the ghost shrimp experienced a beneficial microbiome shift as well, though further research is needed to better understand this possible dynamic, the mechanism behind the increased survival conferred by *Bd* metabolite exposure, and its beneficial effects in other organisms.

Similar results were seen in studies in amphibians, with prophylactic exposure to *Bd* metabolites leading to reduced disease outcomes (*Nordheim et al., 2022*; *Barnett et al., 2023*), which would likely increase survival (*McMahon et al., 2014*). Indeed, tadpoles exposed to *Bd* metabolites experienced more rapid development while still maintaining strong growth and survival (*McMahon, Laggan & Hill, 2019*). This is unexpected, as typically when tadpoles develop faster due to local environmental conditions there is a negative trade off, including smaller size and higher mortality (see *Newman, 1988*; *Bridges, 2002*; *Rohr et al., 2004*). These results support observations in ghost shrimp, and further highlight possible beneficial impacts on survival of exposure to *Bd* metabolites.

We found that *Bd* metabolites did not induce gill recession in ghost shrimp. This finding was important given that previous studies in crayfish have reported gill recession which

was associated with increased mortality (*McMahon et al., 2013*; *Nordheim, Grim & McMahon, 2021*). Importantly, in those previous studies, gill damage was associated with higher doses of *Bd* metabolites; in fact, low doses of *Bd* metabolites did not cause damage in crayfish either. The different responses in gill damage between crayfish and ghost shrimp may be associated with the effects of actual *Bd* infection *vs* exposure to just the *Bd* metabolites. It should be noted that *Bd* positive ghost shrimp have been found in the field (*Rowley, Alford & Skerratt, 2006* but see *Rowley et al., 2007*, as well) and so, it is certainly possible that ghost shrimp are tolerant to *Bd* metabolites. The differential effects in gill damage may also be due to exposure to different specific metabolites as *Bd* produces different metabolites depending on the environment (*Rollins-Smith, Reinert & Burrowes, 2015*). Finally, previous research has found lower concentrations of *Bd* metabolites were not deleterious to other crustaceans (*McMahon et al., 2013*; *Nordheim, Grim & McMahon, 2021*), suggesting different concentrations of *Bd* metabolites may also confer differential effects.

Ghost shrimp exposed to *Bd* metabolites experienced impacts on the cardiac system and had a decreased final heart rate compared to those exposed to ASW. While this finding was based on a 3-s view of the heart rate pre- and post-exposure, the results are interesting, given our observations of increased survival and no gill recession. Some studies have reported cardiac disruptions in amphibians following active *Bd* infections, including decreased stroke volume and delayed cardiac cycles (*Salla et al., 2018*). Additionally, amphibians experiencing effects of chytridiomycosis have impaired epidermal osmoregulation, which was associated with asystolic cardiac arrest (*Voyles et al., 2009*). While we have no evidence that the decreased heart rate is detrimental in the ghost shrimp in our study, and the exact mechanism through which cardiac disruption is precipitated is not known, our findings indicate that the *Bd* metabolites may be associated with altered cardiac function. Further research is needed to fully understand the dynamics associated with *Bd* metabolite exposure on cardiac function because there are likely multiple physiological mechanisms at play, and the impact of *Bd* metabolites on cardiac function may point toward important mechanistic differences between direct exposure to *Bd*, and indirect exposure to *Bd* metabolites.

Ghost shrimp exposed to *Bd* metabolites were observed to spend less time on the dark side of their tank post treatment compared to ASW and were also less likely to exhibit escape behavior in response to an approaching foreign object. Both behaviors deviate from ghost shrimp's expected natural behavioral profile, which include a preference for the dark or covering, and a tendency toward escape, behaviors aimed at avoiding predation. Reduced threat response behaviors, which are likely to increase ghost shrimps' risk of predation, have been linked to disruption of the serotonergic pathway (*Guler & Ford, 2010*; *Fossat et al., 2014*, *2015*). This connection has been supported through research in other decapods, with one study showing that animals exposed to antidepressants such as serotonin reuptake inhibitors, which target the serotonergic pathway, display reduced hiding and escape behavior (*Guler & Ford, 2010*). Indeed, tryptophan, which was identified as one of the most abundantly secreted metabolites in the *Bd* supernatant (*Rollins-Smith, Reinert & Burrowes, 2015*), is an essential amino acid and precursor to serotonin, a

neurotransmitter critical in regulating behavior. Additionally, kynurenine, another *Bd* metabolite, is also central to the serotonergic pathway and immune regulation. Kynurenine, like serotonin, is a metabolite of tryptophan and represents a distinct pathway branchpoint to serotonin, making these two metabolic products key regulators of tryptophan and each other. Importantly, serotonin has been connected to *Bd* susceptibility in amphibians. One study in particular demonstrated correlations between skin serotonin levels and the susceptibility to *Bd*, and perhaps disease progression (*Claytor et al., 2019*). Serotonin was shown to inhibit the growth of *Bd* sporangia *in vitro*, and studies in *Candida albicans* showed that serotonin attenuated virulence in the fungus (*Mayr et al., 2005*). Altogether, this points toward a possible role for *Bd* metabolites, specifically tryptophan, in not only the pathogenesis and virulence of *Bd*, but as prophylactic agents.

This research found that exposure to *Bd* metabolites was not directly deleterious to ghost shrimp, but we did find that these metabolites had potentially adverse impacts that could be detrimental in the field. Given that researchers have been developing a *Bd* metabolite prophylaxis it is crucial that more research be done to understand the impact of these metabolites on critical freshwater organisms. Studies have been mixed in regards to the effects of *Bd* metabolites, with some showing evidence that exposure to high concentrations of *Bd* metabolites can increase mortality (*e.g.*, see *McMahon et al., 2013*; *Nordheim, Grim & McMahon, 2021*) and others finding *Bd* metabolites confer prophylactic effects (*e.g.*, see *Barnett et al., 2023*; *McMahon et al., 2023, 2024*). These discrepancies may be due to differences in concentration of metabolites, or the composition of metabolites organisms are exposed to, which can be dependent on the environment (*Rollins-Smith, Reinert & Burrowes, 2015*). It is also plausible that different strains of *Bd* may have differential impacts on organisms (*e.g.*, see *Barnett et al., 2021*). Different organisms may also display differential responses to *Bd* metabolites, due to inherent physiological differences. It is important that we continue to examine the mechanisms through which *Bd* mediates its impact, including the role of specific metabolites. This is not only important in elucidating the mechanisms through which *Bd* induces deleterious impacts, but also potential targets for prophylactic and therapeutic intervention. The serotonergic pathway may be an important target given existing research showing a key role for serotonin in *Bd* virulence and animal susceptibility, and the identification of tryptophan and kynurenine as *Bd* metabolites. Addressing *Bd's* impact on freshwater organisms may lie in our ability to modulate, at least in part, *Bd's* effects on serotonin and its metabolites.

There have also been few studies focused on understanding the effects of *Bd* metabolites on non-amphibian organisms, despite evidence that they co-occur with amphibians and can be infected by and carry *Bd*. Many freshwater invertebrates, *e.g.*, ghost shrimp and crayfish, could play an important role in understanding the impact of direct and indirect exposure to *Bd* because they are easy to maintain in the lab, their physiology and behavioral responses are well studied, many have transparent bodies and many are not experiencing dramatic declines in the wild. Instead of over utilizing amphibians in this type of work, freshwater invertebrates could be a crucial part of developing and implementing multilayered conservation efforts to protect critically endangered

amphibian populations. Indeed, studying the impact of *Bd* and the metabolites it produces on freshwater invertebrates may provide much needed insight into the mechanisms associated with the pathogenicity and virulence of *Bd* as well as its impact on community ecology.

## CONCLUSIONS

This study examined the effects of *Bd* metabolites in a ghost shrimp model system, focusing on the effects on behavior, cardiac function and mortality. We found that ghost shrimp exposed to *Bd* metabolites experienced reduced mortality when compared to those exposed to a water control and we observed decreased heart rate and reduced threat response behaviors. Previous studies reported increased mortality in crayfish following exposure to high concentrations of *Bd* metabolites, these disparate outcomes underscore the need for more research into understanding the impacts of *Bd* metabolites. Previous research has found that *Bd* metabolites can be used as a prophylaxis for amphibians, however, the use of *Bd* metabolites in a large-scale conservation effort in the field would likely yield exposure of both amphibian and non-amphibian organisms to these prophylactic agents. This work is promising, but it would only be feasible if we continue to do work to understand the impact of *Bd* metabolites on critical non-amphibian organisms.

## ACKNOWLEDGEMENTS

We would like to acknowledge all of the organisms used in this work. We would also like to acknowledge Meiling Bottan and Andrew Bartolomucci for assistance with this experimental work.

### Funding

This work was funded by the National Institutes of Health (Taegan A. McMahon: 1R01GM135935-01) and the Sloan Scholars Mentoring Network Seed Grant (E. Carla Parker-Athill). The funders had no role in study design, data collection and analysis, decision to publish, or preparation of the manuscript.

### Grant Disclosures

The following grant information was disclosed by the authors:
National Institutes of Health: 1R01GM135935-01.
Sloan Scholars Mentoring Network Seed Grant.

### Competing Interests

The authors declare that they have no competing interests.

### Author Contributions

- Ellisa Carla Parker-Athill conceived and designed the experiments, performed the experiments, authored or reviewed drafts of the article, and approved the final draft.

- Liam C. Muldro performed the experiments, authored or reviewed drafts of the article, and approved the final draft.
- Aiza J. Malinias performed the experiments, authored or reviewed drafts of the article, and approved the final draft.
- Taegan A. McMahon conceived and designed the experiments, performed the experiments, analyzed the data, prepared figures and/or tables, authored or reviewed drafts of the article, and approved the final draft.

## Data Availability

The raw data is available in the Supplemental Files.

## Supplemental Information

Supplemental information for this article can be found online at http://dx.doi.org/10.7717/peerj.19815#supplemental-information.

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
