# Peer review of "Exposure to Batrachochytrium dendrobatidis metabolites altered ghost shrimp behavior and reduced mortality"

_PeerJ, doi:10.7717/peerj.19815_

## Round 0.1 · original submission · Major Revisions

The manuscript presents an interesting study on the effects of Batrachochytrium dendrobatidis (Bd) metabolites on ghost shrimp (Palaemonetes paludosus), exploring potential implications for disease mitigation strategies. While the study offers valuable insights, several areas require substantial revision before it can be considered for publication. I agree with the reviewers that the introduction lacks sufficient background on the use of Bd metabolites as a mitigation strategy, which is central to the study’s objective. Strengthening this section will help contextualize the research. Some parts of the Methodology are unclear, for example, information on feeding, the Bd metabolite preparation process, and how behavioral tests were conducted (e.g., details on the light-dark box test). I look forward to reading the revised version of the manuscript.

Reviewer 1 ·

Basic reporting

Overall I think your study is exciting and unique - as you said, there are very few published studies on what Bd does to non-amphibians.

I think this study is valid and interesting, and I am excited to see it published. The writing is clear and confident, and everything is easy to understand and really makes me confident that the work was done to a high level.

The following comments are mostly minor and should be easy fixes but I think they will ultiamtly improve the work and spawn more work like it

Rational: I think it would be useful to explain why you chose this organism and why you chose metabolites rather than actual infection of the zoospores. I would like to understand the rational for why you chose this particular study design. A quick justification in the abstract would be really helpful (and slightly more detail in the intro)
You have done a good job of explaining why metabolites - but did not address why you didn't use active zoospores. Is there a reason you didn’t test it against pathogenic zoospores?
Can you explain why this species of shrimp? Do you know anything about this species and whether they can get infected? Are the metabolite concentrations used here ecologically relevant (would they experience this high a concentration in the wild?)

Methodology/results: The sample size seems to be missing in the methods (or maybe I missed it?) but its not repeated in the results or the figures. Because there were several experiments/analyses done on these shrimp, it would be good to remind the reader how many shrimp you conducted this on. In each section of the methods, and in the figures. and in the results section as well. For example, remind your readers in line 267 what time this is (how much time was analysed and how many shrimp)
It would also be good to put the actually data points on top of your plots and give n = in each of the figure legends.
The results section is okay but it's a little difficult to pick out the important parts. I would suggest reordering it in one of two ways: change the order so that all the significant results are at the top (regardless of the order in the methods) and then list the non-significant results. Or put subheadings that match the methods section, and put the significant results at the top of each section so the key take-home points are clear. Make sure to add effect size to each significant results. Like how much slower was the exposed heart rate at the end of the trial compared both to timepoint 0, and the unexposed on day 8.

For the figures, overlay the actual data points (figures 1 and 2), and add the sample size to the legends (1-4). For figure 3, add error bars - standard error of a proportion. For figure 4, in the legend you should define what the points mean. For the survival analyses I see you actually used a cox regression, which is great, but the figure does not reflect that analysis - its not linear. You should plot it like a survival line , like a Kaplan-Meier curve. It should not be plotted linearly because its not linear, and that's not the model you ran.

Experimental design

See above

Validity of the findings

See above

Additional comments

Minor comments:
Abstract line 48: define ASW
Line 47-49 appears to be word for word from the methodology section (lines 177-180). whats in the brackets doesnt make sense in an abstract.

Bd - is easier to read if you italicize it- I know this is a personal choice, but logically it makes sense to italicize it, and it also makes it a little clearer when reading it that you are referring to a species if its italicized (easier to read and understand)

Lines 79-81: i would suggest rephrasing. it was described recently, but frog declines were noticed well before then - they just didn’t have a cause, partly because disease usually isn’t a cause for such dramatic declines.

Lines 192 and 196, change to superscript (not ^5)

Line 320: wasn't this paper redacted? I think you can explain this with the redaction but keep it in the paper - just be transparent about what has been published.

Reviewer 2 ·

Basic reporting

The reporting of the authors are clear, with sufficient references. There is some duplication, especially with the clarification of abbreviations used, but these can be easily rectified. I agree with the authors that an alternative to amphibians need to be found to use as experimental animal, however, I feel the authors could have elaborated more as to the why ghost shrimp would be a viable alternative to amphibians. Physiological wise, the differences may just be too great. However, if the authors would say that there is some negative influences of Bd on the shrimp, such as reduced reaction to stimuli (danger response) then it could influence the survival of the shrimp as well.

Thus I feel that the author should not try and replace amphibians with shrimp by directly relating the results to the vertebrate group not tested, but rather elaborate on the influence it would have on their host animal, and promote future research trying to find comparative responses between the two types of specimen.

Experimental design

The experimental design seems sound. There are just a few details that may be lacking as addressed in the comments added to the pdf document. These would promote the reproducibility of the experiment by other researchers, and with that in mind the authors may want to add details that would help other researchers perform this experiment, especially since it may become preferrable to use ghost shrimp in stead of amphibians due to ethical considerations. I would also suggest that the authors clarify why they froze and thawed the Bd before filtering, as this would definitely be a question raised. If they can state that the cells were not damaged due to freezing (in which case I do not understand the reason for this step) then it may eliminate uncertainty.

Validity of the findings

No comment

Annotated reviews are not available for download in order to protect the identity of reviewers who chose to remain anonymous.

Reviewer 3 ·

Basic reporting

Parker-Athill et al. studied the effects of exposure to water-soluble chemicals produced by a chytrid fungus, Batrachochytrium dendrobatidis (Bd; a parasite able to cause the disease chytridiomycosis, a disease of conservation significance for numerous susceptible species of amphibians), on a non-amphibian freshwater species, the ghost shrimp Palaemonetes paludosus. More specifically, they investigated effects on cardiac function (heart rate), behavior (activity, escape response, fear-based response), gill tissue recession, and survival. They found (but see my comments on robustness) that ghost shrimp exposed to Bd metabolites experienced decreased heart rate, reduced fear-based behaviors, and reduced mortality compared to those exposed to water control. Those findings are interesting as, on the one hand, they differ from previous research (survival is greater in the survival group here), but on the other hand, the effects on behavior and heart rate might have populational consequences in the wild. Therefore, this paper (i)brings new insights on the benefits and drawbacks of using Bd-metabolites prophylaxis as a mitigation strategy against amphibian chytridiomycosis by testing “side effects” on non-target species, (ii) may stimulate further research on these non-target species, notably with regards to populational impacts of using such metabolites on large scales.
**However, there are a number of problems, in both form and content, that preclude publication in its current state. **
The English is OK but could be more professional and needs some corrections in others. The manuscript (all its sections) can be improved in terms of concision, flow, and clarity. The introduction misses the background on the potential use of Bd metabolites as an in situ mitigation strategy, which is, I think, the overarching goal of this research team. The structure and figures are OK although I have comments (see below), adding sample sizes and measures of uncertainty would be beneficial.
The raw data and metadata are supplied and seem clearly formatted. I confess I did not have time to import this dataset on my end to reproduce the analyses.

Experimental design

This research seems within the scope of PeerJ. The research question is well-defined, relevant, and meaningful, and fills an identified research gap. The investigation, however, could have been more rigorous, in my opinion. It is a pity that the water-soluble chemicals are not identified, as it is known that Bd metabolites can vary depending on the environment. So, the results are difficult to compare with those of previous and future studies. It is also a pity to have to go to the raw data file to know the sample size in each group (n= 19 and 20 for exposed vs. control groups, respectively). Nowhere is this mentioned in the text, if I am not wrong. **The sample sizes are a bit low, for this kind of study species, it could have been greater and the results would have been more robust.** Indeed, all the significant p-values are close to the alpha threshold. Finally, some pieces of information are missing in the materials & methods to correctly understand and reproduce the analysis. It is not clear to me how long the procedures lasted.

Validity of the findings

I think the findings are valid, although, as stated above, I do not have time to reproduce the statistical analyses and more samples would have made the signals clearer. I am a bit skeptical about the heart rate variable, as it has only been assessed for three seconds, which I think is too low to be sure it is representative. I would have done multiple measurements per animal, in one minute, as is done in veterinary medicine. Although heart rate may be much greater in this small species, I still think measuring only 3 seconds is not enough. I encourage the authors to be more concise in the discussion, as the end is quite redundant.

Additional comments

-L38-42: (i) it is somehow implied that Bd is always pathogenic, which is not the case. The fact the Bd can infect asymptomatically some taxa is quite important epidemiologically, as those taxa can form reservoirs or vectors. Please just do not mislead readers into thinking that Bd is systematically causing disease, perhaps add “[…] causing, in susceptible species, disease associated […]. (ii) which Bd hosts are you talking about? Please give precisions.
L45: I am not a fan of the expression “the freshwater keystone species”. Being keystone depends on the context, I mean that a species can be keystone in an ecosystem but detrimental in another (e.g., stoats are important in Europe, but invasive and detrimental in New Zealand where there are introduced). Please think whether this is needed.
L48: please define the acronym ASW
L48-49: the abstract should stand alone, do not refer to the text “below”.
L71-76: please modify the first sentence (the rest is better), it is not the keratin degradation that causes directly lethargy and weight loss. Bd infection, in metamorphosed individuals of susceptible species (this is different for larvae), leads to skin hyperplasia and hyperkeratosis, those lesions cause osmoregulation to fail, which then causes lethargy, cardiac arrest etc. but this is a bit more complicated See also Grogan LF, Skerratt LF, Berger L, et al (2018) Chytridiomycosis causes catastrophic organism-wide metabolic dysregulation including profound failure of cellular energy pathways. Sci Rep 8:8188. https://doi.org/10.1038/s41598-018-26427-z
L79-81: this point (not very well written) is not useful for the paper. Please, delete. Furthermore, Bd have been infecting amphibians in Asia for millions of years, so for this sentence to be meaningful, you should have given this piece of information.
L95: improve the end of the sentence to something like “Bd-metabolite concentrations likely correlate with toxicity”
L100-101: what does mean “pose neutral to beneficial capabilities”?  “have no or even beneficial effects”, if so, reformulate please.
L102: “experienced a prophylactic response”, idem this is awkwardly written, they responded to a prophylactic treatment.
L110: differential  different
L109 and L111: if prophylactic is to be used, say against what infection or disease the prophylactic treatment protects. Note that it can be prophylactic against some parasites but also beneficial on other grounds (increased growth), based on what you describe
L141: it is a shame that the potential usefulness of Bd-metabolite prophylaxis as a mitigation strategy is not mentioned in the introduction, whereas other parts of it are loose and sometimes disconnected to the main topic of the paper.
L143: remove “‘s”
The introduction can be improved in writing and structure. Goals and hypotheses are not given.
L159: fed with what?
The ° in degrees Celsius appears too low throughout the MS.
L166: keep Bd negative and not Bd- throughout the MS. Same for Bd+
L167: is it possible that Bd infection may occur in the digestive tract (as written above) and not on the carapace? In other words, is it likely that you had false negative at the beginning of the experiment?
L182: is a three-second period sufficient to get a representative value of the heart rate? As a vet, it seems to me that it is way too low. Anything can influence the heart rate in such a low period. It would have been better to have measured on longer periods, or done several replicates both pre-exposure and post-exposure and take the average.
L189-199: was the Bd-metabolite preparation prepared only once for all experiments? Since the composition of the Bd metabolite solution is not assessed, can it be expected that the solutions are always the same if prepared at different time points, which would create noise in your results?
L209-213: Could you please explain better how the modified light dark box text works? If in the dark, this is considered to be a normal behavior (of fear)? How long has this been measured? Information is missing on how this was done.
Please review the punctuation : a three second video –> a three-second video, “et al.” should take period
L231: Hyatt et al. (2007) did not propose a qPCR protocol, they tested it for sensitivity and specificity. Boyle et al. (2004) did.
Statistical analysis: please write that you did the analysis in R, with reference and version given. And please cite the creators of packages you used, such glmmTMB.
Experimental procedures and Gill damage: it is not clear how long your experimental procedures last? When and how were the animals euthanized? You must give more information. Were the shrimps placed in groups or alone?
No indication whatsoever on sample sizes ? This could be added to the figures as well.
L245: what is a “post-exposure observation”? this is not defined in the M&Ms, is this the proxy for activity level? This not clear also because you introduced another variable in this activity level, that is, height, I suggest you clarify by making two different paragraph.
Why is there no degrees of freedom indicated for the t statistics?
Figure 4: what does linear mean? Is this the line drawn from the fitted values of the model? Give explanations in the captions.
L302: give a reference
L310: “metabolite related increased survival in ghost shrimp” –>increased survival conferred by Bd metabolite exposure
LL323: cite these previous research papers
L326 (and 336) since in line 333 you do say that you have no evidence that the decreased heart rate is detrimental to shrimp health, you cannot use the word sublethal (which implies toxicity). Modify accordingly. Note that bradycardia may be offset by increased inotropia (which you have not measured), so the cardiovascular may be affected by Bd metabolite exposure, but not necessarily with harmful consequences; arterial pressure should be the measured variable to see if there is any consequence (but I have no idea how to measure it on shrimps). In any case, sublethal is not the appropriate word here.
The decreased flight response may be seen as a sublethal effect however.
L345: reference
L354-355: this type of sentence and syntax is a bit heavy, and too frequent in your MS. Please improve your writing.
L363-364: remove sublet
L366-67: the sentence brings nothing more than the previous one
L370 convey  confer
L372: then it would have been beneficial to the study and this field of research to have analyzed the composition of the metabolite solutions.
L389: it is not clear why you write here that they are invertebrates? Is there a reason pertaining to ethics ? Some crayfish undergo dramatic declines in the wild, at least in Europe because of disease (aphanomycosis) and biological invasions (exotic crayfish)
L377-395: largely redundant with the conclusions. Could be merged with it so as to save words and space.

---

## Round 0.2 · Minor Revisions

I thank the authors for addressing the comments and suggestions of the reviewers. There are just some minor comments pending final approval. I look forward to reading the final manuscript!

Reviewer 1 ·

Basic reporting

The authors addressed all of my comments and the edit looks great. I'm excited to see this in print!

Thanks for adding sample size throughout, its especially helpful in the fig legends (for folks like me who can't always remember what the methods said)

2 small things
Fig 2 - thanks for adding the data points! However you have doubled up on the "outliers" - if you are using ggplot the code you need is geom_boxplot(outlier.shape = NA) - the big black point over the Bd metabolites is duplicated because its normal for boxplots to show the points outside the 95% CI.But when you show all the data points, you need to remove the outliers. Not an issue for Fig 3 becvause there appear to be no outlier points

Figure 4. You need to add error bars. For proportion you should do SE or CI of a proportion. VassarStats used to be able to calculate the upper and lower bar for a proportion though the webpage: http://vassarstats.net/prop1.html

or if you are using R
library(DescTools)
observed = 2 #if the proportion is 2/3
total = 3
BinomCI(observed, total, conf.level = 0.95, method = "clopper-pearson")

Excellent work! Excited for more papers like this published!

Experimental design

See above

Validity of the findings

See above

Additional comments

See above

Reviewer 2 ·

Basic reporting

The basic reporting is still satisfactory, and I believe it was improved with the consideration of the reviewers' comments by the authors.

Experimental design

I am satisfied with the experimental design.

Validity of the findings

The findings of the experiment has been communicated better, and it is easier understood. The authors also acknowledged where the results are perhaps to some counterintuitive, suggesting this as potential future research areas.

Annotated reviews are not available for download in order to protect the identity of reviewers who chose to remain anonymous.

Reviewer 3 ·

Basic reporting

The authors have made substantial revisions and overall, I find the manuscript to be greatly improved. The text is much more precise, concise, and therefore insightful. I thank them for their efforts.

Experimental design

I have only one major comment left. Although this is not the most important variable of this study, I persist in saying the design regarding the heart rate measurement was not appropriate and that the inherent results should not be trusted. That the authors started with 3s-period and kept using this period to be consistent does not make the method any more valid. This is not enough time to get a meaningful and representative value of the heart rate at rest. Also, in the rebuttal, authors state that the suggested period of one minute may lead to more bias because of the restraining: (i) I disagree, wild animal vets and vets in general often wait for the animal to calm down after restraint to measure heart rate at rest (stress, through hormones like catecholamines and corticosteroids, greatly increases heart rate on very short periods), and the longer the period, the more chance the animal can calm down; (ii) there is no mention of containment in your method section. If you did use containment/restraint, state how. This is indeed important, because, as said, the restraint method and the stress it induces can greatly alter physiological variables such as heart rate. The reader should know that, I thought that heart rate measurement was performed without disturbing the shrimp (as for the behavioral monitoring); (iii) whatever the chosen period, you should have made multiple measurements, not just once at the beginning and one at the end.

Validity of the findings

In conclusion, I don’t believe this variable (heart rate) should be trusted. If it is to be kept (I would not keep it), I would like the authors to be extremely transparent and state that the results regarding this variable are highly uncertain and only indicative.
That said, this variable is not the most important one in this study, and even if it is invalid, this does not change other results, which are insightful and worthy of publication.

Additional comments

Some details regarding the statistical analysis are missing (like the link functions of the GLMs).
Below are some minor comments:
L39: I suggest you place “susceptible” (same L65) not in this sentence but in the following one, for two reasons: (i) Bd has been indirectly implicated in declines of non-host species, such as snakes predators of amphibians (Zipkin et al. 2020), which are non-susceptible; (ii) Bd can infect host without causing disease, for instance in reservoir amphibian species like bullfrogs or African clawed frog, therefore it make more sense to write L40 “Bd can infect multiple taxa, causing disease in susceptible species […]”.
L45: just a remark, my point in the previous review was not to make you not use the word “keystone”, but to put it into more context, which you did in the introduction by stating gosht shrimp roles, but perhaps this could be a little bit more explicit in the abstract in room is available. You can use “keystone” l57, as I think this is an important word.
L57: remove comma
L60 61: true but this is not the main take-home message in my opinion. What you added L56 is more appropriate as the main conclusion (it better illustrates the significance of your work).
L68 “vectored” is not the appropriate word in this context. Please use “transmitted”.
L75: “pathologies” is also not the right word here. What you mean is “lesions” (which I recommend) or “pathological features”. Strictly speaking, pathology is the science that studies disease. More broadly (but some do not like it), this word is used as a synonym of disease. In any case, there is a difference between disease (broader) and lesions (localized): a lesion can be a sign of a disease, a disease may cause lesions. Same L87, I would use “disorders”.
L107 “against Bd infection”, I agree but would even expand this statement: prophylaxis may preclude infection, and/or it may also reduce infection intensity (parasite burden), which has been shown (in some susceptible species only) to correlate with disease severity and outcome.
L161-62: “impacts the freshwater community”: I would even use “freshwater ecosystems” since you used a model species with important roles, which you highlighted in the previous §, in ecosystem functioning.
L253: Byole  Boyle
L270: “number of observations post-exposure”, thank you for answering my question in the previous round of review, but while I do understand what you mean by that now, I don’t think a reader will even with your additions elsewhere. I would encourage you to be as explicit as possible when you describe covariates of a model.
L272 onwards: remove the mentions of function summary. What we want to know is, most of all, the link function you used in the model you fitted (identity for LM and LMM, Poisson, Negative binomial, Beta, etc for GLM/GLMM). Indicate whether you specified other parameters in your model too. One needs to be able to reproduce your results. Without the link function, it is not possible
Figure 1: add the units (days) for time on the x-axis. So was there no mortality from day 6 to14 (last mortality on day 5)? Why is that?
L296: give the confidence intervals
Figure 2: if you talk about gill recession in the figure title, show the data. Also, there seems to be both the data points (jittered) and also a point (thicker) representing an outlier. If you put jittered points to show the data (as asked by another reviewer, and I confirm it is useful), remove the outlier.
Figure 3: the title of the y axis should be “average percent/proportion of time …”

L337-43: I find that, in the first paragraph of the Discussion, there is too much text on effects of Bd metabolites on tadpoles (another study, but still 7 lines of text). The first paragraph of the discussion should really highlight your main findings and what they mean. Then you can go in more details.
L347: please remove “certainly” if you have no data here
L354-55: I don’t understand why you compare with studies on crayfish challenged by live Bd, since here shrimps were challenged only metabolites.
L420: “may lie in or ability to modulate”  may lie in our ability to modulate?
L421: “effects of this pathway” is a bit vague? Not sure what you mean by that.

---

## Round 0.3 · accepted · Accept

thank you to the authors for addressing all the comments and suggestions from the reviewers. I look forward to reading the final published version!

Reviewer 3 ·

Basic reporting

The authors addressed my comments and I have only a few remarks, but nothing that necessitates another round of review. It is good for publication.

For Fig.2 legends, I do understand why authors wanted the "did not induce gill recession" to appear somewhere (here, a header, which I called a figure title), but I find this inappropriate in this case since there are no data presented on gills in this figure. I leave this to the Editor, however, this is a simple figure, and authors could add another facet for gill, on top of the other for heart rate. This will not increase the number of figures.

Experimental design

I encourage authors to consider making several measurements of heart rate both prior to and after exposure, in case they want to investigate the effect on cardiovascular function in future studies.

Validity of the findings

noting to add

Additional comments

Thank you for this work.